# Protective Role and Functional Engineering of Neuropeptides in Depression and Anxiety: An Overview

**DOI:** 10.3390/bioengineering10020258

**Published:** 2023-02-16

**Authors:** Nathalie Okdeh, Georges Mahfouz, Julien Harb, Jean-Marc Sabatier, Rabih Roufayel, Eddie Gazo Hanna, Hervé Kovacic, Ziad Fajloun

**Affiliations:** 1Department of Biology, Faculty of Sciences 3, Campus Michel Slayman Ras Maska, Lebanese University, Tripoli 1352, Lebanon; 2Department of Psychology, Faculty of Arts and Sciences, Beirut Campus, American University of Beirut, Beirut P.O. Box 11-0236, Lebanon; 3Faculty of Medicine and Medical Sciences, Dekouene Campus, University of Balamand, Sin El Fil 55251, Lebanon; 4CNRS, INP, Inst Neurophysiopathol, Aix-Marseille Université, 13385 Marseille, France; 5College of Engineering and Technology, American University of the Middle East, Egaila 54200, Kuwait; 6Laboratory of Applied Biotechnology (LBA3B), Azm Center for Research in Biotechnology and Its Applications, EDST, Lebanese University, Tripoli 1300, Lebanon

**Keywords:** neuropeptides, depression, anxiety, gut–brain peptide, opioid peptide, pituitary hormone peptide

## Abstract

Behavioral disorders, such as anxiety and depression, are prevalent globally and touch children and adults on a regular basis. Therefore, it is critical to comprehend how these disorders are affected. It has been demonstrated that neuropeptides can influence behavior, emotional reactions, and behavioral disorders. This review highlights the majority of the findings demonstrating neuropeptides’ behavioral role and functional engineering in depression and anxiety. Gut–brain peptides, hypothalamic releasing hormone peptides, opioid peptides, and pituitary hormone peptides are the four major groups of neuropeptides discussed. Some neuropeptides appear to promote depression and anxiety-like symptoms, whereas others seem to reduce it, all depending on the receptors they are acting on and on the brain region they are localized in. The data supplied here are an excellent starting point for future therapy interventions aimed at treating anxiety and depression.

## 1. Introduction

Youth and adults who suffer from mental and behavioral disorders such as depression and anxiety place a heavy strain on themselves. These disorders jeopardize the health and wellbeing of those who suffer from them. In fact, a behavioral disorder affects 14–20% of young people at some point in their lives [1].

Depression is a complex behavioral disorder that largely affects populations globally, with a higher prevalence in women than in men [2]. Depression has recently been on the rise as a variety of economic and environmental elements wear down globally. According to the World Health Organization (WHO), as of 2020, over 300 million people suffer from depression [3]. Furthermore, as stated by the WHO, depression will become one of the leading causes of disability worldwide by 2030 [4]. 

Anxiety disorders are among the most common classes of psychiatric disorder in most countries of the world [5]. Anxiety disorders are characterized by a wide range of physical and emotional symptoms, as well as shifts in behavior and cognition. These conditions are among the most common psychiatric disorders that can lead to excessive morbidity, the need for medical care, and functional impairment. Similar to depression, women are more likely than males to experience anxiety disorders [5]. In fact, due to their common comorbidities and risk factors, anxiety and depressive illnesses are intertwined. Despite the fact that some anxiety disorders have little to no correlation with depressive disorders, clinical and epidemiological research demonstrates that these disorders are rarely present separately [6].

Numerous studies, which are covered in this review, have demonstrated how neuropeptides contribute to the onset and severity of behavioral disorders such as anxiety and depression. Neuropeptides have been shown to have direct and indirect effects on the pathophysiology of behavioral disorders such as depression and anxiety. However, their effect depends on the targeted area. For example, one neuropeptide can induce a behavioral disorder in one area, but reduce depressive behavior in another brain area.

Peptides are largely spread in the key regions of the brain associated with emotional regulation. Peptides also have a significant modulatory role in the monoamine systems, particularly in conditions where the nervous system is under stress, challenged, or afflicted with a disease, such as in the case of behavioral disorders [7]. The signaling molecules in the brain that are by far the most numerous and varied are called neuropeptides. Numerous physiological roles and processes that neuropeptides take part in allow us to identify their modes of action [8]. 

The primary focus of this review is on research demonstrating the behavioral impact of neuropeptides on behavioral disorders, particularly depression and anxiety, in different animal models, as well as human patients.

## 2. Neuropathology of Depression and Anxiety

### 2.1. Depression 

It has previously been shown that people with depression experience significant alterations in their neuroanatomy. The main change observed in depressed patients was a reduction in overall brain size, specifically in the ventromedial prefrontal cortex, and notably in the left anterior cingulate and orbitofrontal cortex. The lateral prefrontal cortex, hippocampus, and striatum also showed moderate volume reductions [9]. Moreover, abnormal activity was noted in lateral, frontal, and temporal cortices, as well as the insula and the cerebellum. These brain regions showed a lack of activation during negative emotions induction [10]. Additionally, numerous studies have discovered that different neuropeptides play important roles in depression, with these roles varying depending on the region where they are operating. 

### 2.2. Anxiety 

The amygdala, which plays a critical role in organizing the automatic threat response and integrating information gathered from various sensory organs, is the principal area of the brain associated with anxiety. Prior conducted neuroimaging studies have shown an activation of the amygdala, in both human and animal models, following a fear stimulus [11]. It has been hypothesized that amygdala malfunction may contribute to emotional dysregulation, which is the primary symptom of anxiety disorders [12]. Numerous studies demonstrating the involvement of multiple neuropeptides that seem to have anxiolytic effects, have shown the neurochemical circuitry underlying anxiety disorders. The detailed effect of each neuropeptide on different animal and human models is presented in the review and summarized in Table 1.

## 3. Preclinical Models in Depression and Anxiety

Various models are used to induce depressive and anxious behavior in animals. The preclinical models used in the studies cited in this review are described in this section. Each model used in each study is described in detail in Table 1.

### 3.1. Vogel’s Conflict Test (VCT) 

The conflict test known as the Vogel test is now the industry standard for quickly evaluating a drug’s potential anxiolytic effect. This method severely reduces the animal’s water consumption by using mild electrical shocks to discourage the drinking behavior. In this experiment, rats are denied access to water for 48 h before a 3 min session in which drinking is sanctioned by delivering a little unpleasant shock through the bottle spout after 20 licks. As a result, it can be determined that a certain drug’s ability to make users absorb more shocks, akin to drinking more water, is an indicator of its anxiolytic properties [40].

### 3.2. Elevated Plus Maze (EPM)

One of the tests that is most frequently used to assess anxiety-like behavior is the EPM test. The experiment takes advantage of mice’s innate fear of high, open areas as well as their intrinsic curiosity about unfamiliar environments. The apparatus has a central area with open and closed arms that are crossed perpendicularly in the middle. The mice have access to each arm and can roam freely between them. The number of entries and the length of time mice spend in open arms are signs of anxiety brought on by open spaces [41].

### 3.3. Open Field Test (OFT, OPF)

The OFT, a popular model of anxiety-like behavior developed to assess emotional responses in animals, places an animal in an unfamiliar environment from which it is unable to escape due to surrounding barriers. The animal is placed at the center or close to the walls of a device (circular, square, or rectangular), and numerous behavioral traits (typically grooming and horizontal/vertical activity) are watched over for a set amount of time. Because this is an unusual and possibly pressure field for the animal, rats frequently avoid the illuminated center of the device and prefer to stay close to the walls [42].

### 3.4. Light–Dark Box (LBD) Test

The LBD test is based on rodents’ intrinsic aversion to areas with bright lighting and their innate exploratory behavior in response to small stressors such as light and novel environments. The test apparatus consists of a large, illuminated unpleasant compartment (two-thirds) and a small, dark safe compartment (one-third). Male mice were used in the test’s development. Age, weight, and strain may be important variables. The baseline level in the control group determines how well an anxiolytic drug can encourage exploratory activity. External stressors of varying types and intensities could be to blame for the disparities in results reported by different laboratories. The LBD test may be helpful in predicting whether a mouse will exhibit anxiolytic or anxiogenic behavior [43].

### 3.5. Single Prolonged Stress (SPS)

SPS is a procedure that exposes the animal to traumatic levels of stress. It entails sequential exposure to three stressors over the course of one continuous session. This strategy was initially designed to induce a significant stress reaction through three different mechanisms: constraint, forced group swimming, and medication (ether). Each session consists of 2 h of restraint, a 20 min group swim, and ether exposure until loss of consciousness. This incubation period, also known as a sensitization period, lasts 7 days on average. This procedure induces post-traumatic stress disorder (PTSD)-like symptoms such as depression and anxiety [44].

### 3.6. Olfactory Bulbectomy (OB)

In this technique, mice’s olfactory bulbs are removed, causing neuronal rearrangement and the expression of changes on many levels, including behavioral, neurochemical, immunological, and neuroendocrine abnormalities that are comparable to those seen in depressive disorders. As a result, this model is frequently used to investigate the neurological bases for the pathophysiology of depression and to evaluate antidepressant drugs. The most common modification seen in the olfactory bulbectomized mouse model when it is subjected to a new stressful setting is hyperactivity. Selectively reducing this behavioral reaction is one of the effects of chronic antidepressants [45].

### 3.7. Forced Swim Test (FST) or Porsolt Swim Test (PST)

The FST, often referred to as the PST, is a mouse behavioral test that is employed to evaluate the efficacy of novel treatments for depression, as well as experimental procedures designed to induce or prevent depressive-like states. Mice are placed in a solid, clear tank that is filled with water, and the movement behavior associated with their attempts to escape is observed. The forced swim test is simple to administer accurately and requires little specialized apparatus. The FST must be carried out according to specified procedural guidelines, and any unnecessary stress on the mice must be minimized [46].

### 3.8. Flinder’s Sensitive Line (FSL) 

The FSL strain of rats was created as a line with extreme behavioral sensitivity to the cholinesterase inhibitor di-isopropyl fluorophosphate. It is employed as a rat model of depression because it exhibits several traits that are similar to those of human depression, including increased rapid eye movement (REM) sleep and reduced mobility in swim test that can be lessened by long-term antidepressant therapy [47].

### 3.9. Tail Suspension Test (TST) 

The TST is a mouse behavioral test that can be used to evaluate treatments that are expected to have an effect on depression-related behaviors, as well as potential antidepressant medications. In this procedure, tape is used to suspend mice by their tails so they cannot run away or grab onto nearby objects. This test typically lasts 6 min and measures the ensuing escape-oriented behaviors [48]. 

## 4. Neuropeptides

A peptide is a chain of amino acids, with a maximum length of 50 amino acids, interconnected by bonds known as peptide bonds. Peptides are distinguished from proteins primarily by their shorter length. When a peptide is released by a neuron, it is referred to as a neuropeptide. In fact, a neuropeptide, by definition, is a short-chain amino acid that is secreted by neurons and acts on neural substrates [8]. According to H. Burbach, the hallmarks of neuropeptides can be narrowed to three: (1) their genes and proteins are both expressed and synthetized by neurons, (2) they are stored in vesicles within the neurons and released when required, and (3) they are able to modulate, through the neurons’ receptors, the actions and functions of these neurons [49]. 

Neuropeptide biogenesis is relatively simple. All neuropeptides derive from the pro-peptide, a precursor protein, which goes through a C-terminal amidation in the endoplasmic reticulum. While passing through the endoplasmic reticulum (ER)–Golgi route, the pro-peptide undergoes more modifications such as glycosylation. The precursor of neuropeptides is then stored in the compartment trans of the Golgi apparatus within vesicles. These vesicles subsequently become acidic, activating proteolytic enzymes that cause the precursor to mature into a neuropeptide. The now mature neuropeptides are stored in a dense vesicle ready for release when stimulated. Neuropeptides do not go through presynaptic reuptake, in contrast to monoamines and amino acids. Their breakdown by extracellular peptidases such as neutral endopeptidase and angiotensin-converting enzyme puts a stop to their synaptic activities. The catabolic byproducts of the released peptide may serve as signals in a number of peptidergic systems, both presynaptically and in the target neuron [49]. 

Neuropeptides secreted by neurons have the ability to affect both neuronal and non-neuronal cells. In the human brain, numerous neuropeptides have been identified, each of which can affect the physiology of neurons and modulate gene expression in seconds or even hours and days. In addition to the brain, neuropeptides also have significant impacts on the body’s neurological systems. Their biological activities can be observed at the biochemical, genetic, behavioral, cellular, and organismal levels [49]. 

Additionally, there are numerous varieties of neuropeptides that fall into the following categories: (1) gut–brain peptides, (2) hypothalamic releasing hormones peptides, (3) opioid peptides, (4) pituitary hormones peptides, and (5) miscellaneous peptides. The roles of each of these classes of peptides in behavioral disorders, notably anxiety and depression, are discussed in this review.

### 4.1. Gut–Brain Peptides

The term “gut–brain” refers to the bidirectional axis connection between the gut and the brain (Figure 1). By way of neuroendocrine substances originating from the adrenal cortex and medulla, this axis plays a part in the continuing communication between the gut and the brain [50]. Neuropeptides such as neuropeptide Y, substance P, neurotensin, and galanin are expressed at all levels of the gut–brain axis. 

These neuropeptides are essential for the bidirectional transmission between the gut and the brain. In theory, certain microorganisms can make compounds such as neuropeptides, and the gut microbiota reacts to these molecules and to gut hormones if the appropriate receptors are expressed [50]. In recent years, increasing evidence has suggested that the microbiota–gut–brain axis plays a key role in regulating brain functions, particularly emotional processing and behavior. Numerous studies, covered in this review, have demonstrated that behavioral disorders, notably anxiety and depression, are significantly influenced by neuropeptides of the gut–brain axis.

#### 4.1.1. Neuropeptide Y (NPY)

NPY is a neuropeptide of the gut–brain axis constituting 36 amino acids, rich in tyrosine residues, with a molecular weight of 4272 Da. It is abundant in both the central and the peripheral nervous systems [51]. NPY plays a role in regulating various systems throughout the body such as memory, anxiety, fear, and stress [52]. The cerebral cortex, locus coeruleus, hippocampus, brainstem, and hypothalamus are the major sites of NPY production in the brain. The hypothalamus has the highest concentration of NPY.

The physiological effects of NPY are mediated by G-protein-coupled receptors. Five of these receptors are designated as Y1, Y2, Y4, Y5, and Y6 receptors [53]. Among these receptors, Y1, Y2, and Y5 are dominant within the central nervous system (CNS). Gi/o links NPY receptors to a number of downstream pathways, including those that block adenylyl cyclase, control intracellular Ca^2+^, activate MAPK, stimulate inward rectifying potassium channels, and trigger Ih currents.

Food intake [54], sexual behavior [55], information processing [56], cognition, learning, and memory [57], as well as stress and anxiety [57], are all significantly regulated by NPY and NPY receptors. Furthermore, a growing body of evidence has shown the antidepressant and anxiolytic effects of NPY. 

NPY in Depression

It has been shown that depressed patients’ plasma [58] and cerebrospinal fluid [59] levels of NPY are reduced. Additionally, depressed suicide patients had lower levels of NPY mRNA and higher levels of NPY1R and NPY2R mRNA in the prefrontal cortex (PFC) and hippocampus areas [60]. Moreover, a study conducted on 34 adult patients with epilepsy undergoing a temporal lobe surgery for seizure control showed a significant positive correlation of the density of NPY-positive neurons in the basolateral amygdala with depression scores [61]. Furthermore, treatment with antidepressants, specifically venlafaxine and escitalopram, showed increased normalized serum levels of NPY after 8 weeks in depressed patients in comparison to healthy controls, in a study that included 40 patients with depressive and anxiety symptoms and 32 healthy controls. However, following therapy with sertraline and fluoxetine, there were no noticeable changes in NPY levels [62]. Therefore, current research has shown that NPY has an antidepressant impact. Depressive-like behavior in a rat SPS model of PTSD diminished following NPY intranasal treatment [63]. In the same animal model, another study showed that administration pre-SPS of NPY with HS014, an MC4R (melanocortin four receptor) antagonist, at low doses, averted the development of depressive-like behavior. Furthermore, the administration of NPY and HS014 post SPS reduced the FST behavior of “giving up”, a sign of depressive-like behavior [64]. Another research study using the SPS rat model of PTSD revealed that intranasal injection of the NPY Y1R agonist [D-His26] NPY stopped depressive-like behavior from emerging. However, intranasal administration of the Y2R agonist (NPY 3-36) was unable to prevent this behavior. Therefore, it is possible that [D-His26]NPY activation of Y1R is sufficient to control or stop depressed behavior [65].

Similarly, a study using the FSL rat model of depression revealed that the antidepressant treatments employed in the PST, including running and the combination therapy with escitalopram and a running wheel, significantly increased the NPY mRNA. Running increased NPY mRNA expression in all examined hippocampus regions, whereas the combined treatment increased NPY mRNA expression in the hilus and dentate gyrus. This suggests an antidepressant role of NPY [66]. A study on mice used a stress-induced depression-like paradigm to demonstrate this function of NPY. Following exposure to stressors, these mice received Kososan for 28 days, an herbal antidepressant. The Kososan administration was then followed by intracerebroventricular (i.c.v) injection of BIBO3304, which is an antagonist for the neuropeptide Y1 receptor (NPY1R), for 7 days. This resulted in Kososan’s antidepressant effect being blocked, indicating that Kososan acts on the NPY1R to modulate its antidepressant effect [67]. Studies also demonstrated the antidepressant effect of NPY on human models. A study of 256 depressed patients found that the 399C allele of the NPY SNP rs16147 was associated with a decreased ability to respond to treatment, with a slower initial response after 2 weeks and lower remission rates after 4 weeks. These findings back up the idea that the functional 399C/T polymorphism (rs16147) may play a role in modulating the antidepressant treatment response in patients with anxious depression [68].

NPY in Anxiety

The involvement of NPY in anxiety disorders has been proven in multiple recent studies. Serum NPY levels were found to be lower in patients with anxiety and depression compared to healthy controls. Additionally, a study conducted on 616 adults that faced the 2004 Hurricane in Florida revealed that the rs16147 NPY gene, a functional single-nucleotide polymorphism in the promoter region of NPY, was linked to an elevated risk of a diagnosis with generalized anxiety disorder (GAD) with the condition of having been highly exposed to the hurricane [60]. Additionally, a study on two strains of zebrafish (NPY-KO7 and NPY-KO11) with deletions of seven and 11 nucleotides in NPY, respectively, revealed that, under extreme stress, the NPY gene-deficient strains (NPY-KO) displayed anxious behavior such as reduced movement, freezing, and swimming on the tank’s edge in comparison to wildtype fish. Moreover, levels of genes associated with anxiety were notably higher in the NPY-KO strain than in the wildtype fish. These findings indicate that a deficiency in the NPY gene induces anxiety-like behavior in zebrafish [69]. This severe anxious behavior after acute stress exposure in NPY-deficient zebrafish was reduced by treatment with Ninjinyoeito, a Japanese Kampo medicine that activates NPY neurons. However, its anxiolytic effects were probably brought about by the deactivation of noradrenaline neurons [70]. 

The role of the NPY system in the anxiolytic action of allopregnanolone (ALLO), a steroid associated with the regulation of anxiety, was demonstrated in a mouse model where the VCT was utilized to quantify the anxious behavior by counting the number of shocks. The bilateral administration of [Leu31, Pro34]-NPY, NPY1/Y5 receptor agonists, into the central nucleus of amygdala (CeA), elevated the number of shocks in VCT, which is an anxiolytic-like effect, whereas the administration of NPY1R antagonist BIBP3226 caused anxious response. These findings suggest that the NPY1R is involved in the NPY anxiolytic effect. Furthermore, the treatment with NPY or [Leu31, Pro34]-NPY and ALLO increased the number of shocks in VCT, indicating an anxiolytic-like effect. However, treatment with BIBP3226 and ALLO decreased the anxiety-reducing effect of ALLO suggesting that the anxiolytic effect of ALLO is mediated by acting on the NPY system via the NPY1R in some areas of the brain [71]. In accordance with these findings, NPY has been found to mediate the anxiolytic effect of agmatine, a neurotransmitter involved in regulating anxiety, in rats. This study found that giving NPY or [Leu31, Pro34]-NPY intravenously increased the anxiolytic effect of agmatine in the mice’s VCT. BIBP3226, on the other hand, reduced the anxiolytic effect of agmatine. This suggests that agmatine regulates anxiety in the CeA via the NPY system [72]. The anxiolytic effect of NPY was also demonstrated in a mouse study in which the ablation of NPY neurons in the nucleus accumbens (NAc) increased anxious behavior while activation of these neurons increased anxiolytic-like behavior [73]. 

#### 4.1.2. Substance P (SP)

SP is an 11-amino-acid undecapeptide neuropeptide belonging to the neuropeptide family of tachykinin and largely distributed in both the central and the peripheral nervous system [74]. There are three NK receptors, the neurokinin 1 receptor (NK1), NK2, and NK3 [75]. SP acts through its preferred receptor, neurokinin type 1 (NK-1R), which is a transmembrane receptor bound on various body cell types including white blood cells, endothelia of the blood vessels, fibroblasts, and neurons. Through altering cellular signaling pathways, SP functions in the brain as a neurotransmitter and/or neuromodulator. SP plays an important role in many functions such as memory processes, vasodilation, cell growth and proliferation, and behavior modulation. Furthermore, SP acts via G-protein-coupled receptors, which are abundantly expressed in brain regions responsible for behavior regulation, and it acts via inositol trisphosphate/diacylglycerol (IP3/DAG) and cyclic adenosine monophosphate (cAMP) depending on the cell type [74]. 

SP in Depression

Recent research has demonstrated SP’s role in the pathogenesis of depression. A total of 91 stroke patients were included in a study, and they were split into PS and PSD (post stroke with depression and post stroke without depression). SP plasma and cerebrospinal fluid levels were much higher in PSD patients than in PS patients. The results also showed a positive correlation between the levels of SP and the degree of depression [76]. Furthermore, a study conducted on two rat models of depression, one exposed to chronic mild stress (CMS) and the other chronically administered the antidepressant clomipramine (CLI), showed that the microinjection of SP receptor antagonist (SPA) into the lateral habenula (LHb) lessened the time of immobility and elevated the time of climbing. These findings suggest that the SPA has an antidepressant effect mediated by the LHb [77]. Moreover, a study on olfactory bulbectomized mice showed that the deletion of NK1 receptor lessened the OB-triggered elevation in exploratory behavior and locomotor activity. Furthermore, the NK1 deletion reduced the OB-induced changes in serotonin within the amygdala. These findings indicate the antidepressant role of NK1 following bulbectomy [78]. 

SP in Anxiety

A large number of studies have demonstrated the role of SP in the pathogenesis of anxiety and anxious disorders. According to a study on rats using the EPM model, animals given intra-cerebrospinal injections of the selective NK-1 receptor antagonist L822429 displayed noticeably higher rates of entry into open arms and were more likely to remain in them for longer periods of time than unaffected controls. These findings suggest the anxiolytic effect of the antagonist of the NK-1 receptor [79]. This is in accordance with a study conducted on EPM model of rats that showed the anxiogenic effect of SP via its NK-1 receptor. This study demonstrated that the injection of SP into the central nucleus (CeA) and the medial nucleus of the amygdala (MeA), but not into the basolateral nucleus of the amygdala, caused anxiogenic-like effects. These findings were confirmed by injecting Sar-MetSP, a neurokinin agonist with high affinity for the NK1 receptor in brain tissue. This indicates that the SP neuropeptide may cause the effects of reducing anxiety via the activation of the NK-1 receptor in the MeA and the CeA [80]. Furthermore, when injected to the dorsal hippocampus (DH), SP presents anxiolytic-like effects, whereas the injection of SP into the ventral part of the hippocampus showed no modification of behavior [81].

#### 4.1.3. Neurotensin (NT)

NT is a neuropeptide made up of 13 amino acids and was first isolated from the hypothalamus of a bovine. NT is largely distributed through the central nervous system. This neuropeptide has been implicated in various physiological functions such as pain, reward, appetite, memory, and behavioral processes. Neurotensin acts by interacting with specific receptors NTS1, NTS2, and NTS3. NTS1 and NTS2 are G-protein-coupled receptors, whereas the NTS3 is a single transmembrane receptor. NT is highly selective for the NTS1 [33]. This receptor has been at the center of interest of several studies targeting the treatment of mental disorders as it has an influence on the monoamine neurotransmitter system and produces antipsychotic and anxiolytic effects [82]. 

NT in Depression

The effect of NT was shown in a study conducted on 160 men and women suffering from obesity with symptoms of depression and anxiety by studying the correlation between these symptoms and levels of NT and xenin, which is an anorexigenic neuropeptide. This study indicated that both neuropeptides were positively correlated with stress, anxiety, and depression in women but not in males. This suggests a sex-specific association of NT with the pathophysiology of depression and anxiety [83]. Furthermore, the effect of the NST1 receptor agonist (PD149163) was shown on a FST model of mice. The intraventral tegmental administration of PD149163 caused an antidepressant-like effect in the forced swim test. An animal model with the antidepressant imipramine was used to further demonstrate the NST1 receptor agonist’s function. In this model, the agonist also displayed an anti-depressive-like effect [82]. 

Moreover, the implication of NT on depressive symptoms was studied on an animal model, emphasizing gene–environment interactions. This study took two animal models, the FSL rat model of depression and FRL, subjected to maternal separation and studied the effect on NT-like immune-reactivity (NT-LI). The results noted that FSL rats had higher baseline concentration of NT-LI than FRL in the periaqueductal gray and temporal cortex and had notably different reactions to maternal separation. Furthermore, FRL rats exhibited an increase in NT-LI concentration in the periaqueductal area following maternal separation. In FSL rats, however, NT-LI concentrations increased in the hippocampus, nucleus accumbens, and entorhinal cortex, whereas a decrease was noted in the temporal cortex after maternal separation. These findings imply that elevated NT levels are among the biological correlates of depression, and that the stress of mother separation would cause additional neurotensin changes in specific brain regions in genetically susceptible individuals [84].

NT in Anxiety

The effect of NT on anxiety has been shown in recent research. A study conducted on the EPM rat model of anxiety showed that the bilateral microinjection of NT into the ventral pallidum had an anxiolytic-like effect at the dose of 100 ng but no effect was noted at the dose of 250 ng. Furthermore, in an OPF rat model of anxiety, they injected 35 ng of an NT1R SR 48,692 alone or 15 min prior to NT treatment. The results showed that the antagonist alone had no effect; however, when injected before the treatment with NT, the antagonist inhibited the effect of neurotensin. These findings suggest that NT has an anxiolytic effect acting through its NT1 receptor [32]. Similar results were observed in another study conducted on an EPM model of rats [85]. Moreover, the effect of the neurotensin receptor 1 was further studied in research on an EPM, LBD, and OF anxiety model of rats. This study showed that the injection of a NTS1 agonist or NT into the prelimbic region of medial prefrontal cortex (PrL) induced anxiogenic-like effects. In contrast, the injection of an NTS1 antagonist into the PrL had no anxiety-like effect on normal rats but reduced the anxiogenic stress-related effects. This study also showed that the downregulation of NTS1 in the PrL caused anti-anxiety-like effects in stressed rats. These findings imply that NTS1 in the PrL plays a role in anxiety regulation [33]. 

#### 4.1.4. Galanin (GAL)

GAL is a 29–30-amino-acid neuropeptide distributed in the central nervous system of humans and other mammals [86]. GAL appears to play a significant role in the neurobiology of mood disorders, as shown by its colocalization with serotonin in the DRN and with noradrenaline in the locus coeruleus [19]. GAL acts through the activation of GAL1, GAL2, and GAL3 metabotropic receptors that are largely found in the brain of rats. GAL1 and GAL3 have inhibitory effects by acting on a Gi protein inducing a K^+^ efflux, thus reducing neurotransmitter release, whereas GAL2 acts by activating a Gq protein, causing an increase in intracellular Ca^2+^ concentration and, therefore, increasing neurotransmitter release mediated by GAL [87]. Hence, the effect of GAL in brain regions depends on the distribution of these receptors. GALR1 is majorly expressed in the locus coeruleus, hypothalamus, ventral hippocampus, and nucleus accumbens. GALR2 is expressed in the hypothalamus and in the limbic system [88]. Contrarily, the hypothalamic–pituitary (HP) axis expresses GALR3 abundantly while the brain expresses it in low levels [89].

GAL in Depression

The effect of GAL on depressive-like symptoms highly depends on the receptor it is acting on. A study conducted on an FST rat model of depression showed that exposure to stressors induce the release of GAL but the effect on its receptors varies. In this study, the i.c.v administration of GAL, GALR1 agonist M617, or GALR2 antagonist M871 increased immobility time, whereas the i.c.v. administration of GalR2(R3) agonist AR-M1896 decreased it. These findings suggest that GALR1 mediates the pro-depressive effect of GAL, while GALR2 mediates its anti-depressive effect. [19]. On the basis of the findings of this study, a subsequent study using the FST rat model of depression revealed that the intra-DRN injection of galanin and a GAL2 agonist (AR-M1896) resulted in an antidepressant effect during the forced swim test, whereas the intra-DRN administration of a GAL1 agonist (M617) had no effect. Additionally, the OFT rats’ locomotor activity was unaffected by the intra-DRN delivery of AR-M1896 and M617.Furthermore, an intra-DRN pretreatment with the selective GALR2 antagonist M871 reduced the antidepressant effect of GAL. These findings suggest that galanin exerts an antidepressant-like effect in the dorsal raphe nucleus via its GAL2 receptors [20]. 

In accordance with these findings, a study conducted on OB rats showed that the i.c.v administration of GAL (1-15) improved the antidepressant effect of escitalopram, whereas the GALR2 antagonist M871 blocked this effect [21]. Another GalR2-selective agonist, M1160, was shown to have a similar antidepressant effect to the antidepressant medication imipramine after i.c.v administration in a tail suspension test in mice [22]. Furthermore, the i.c.v infusion of galanin in rats enhanced the NPYY1R agonist [I125]-[Leu31, Pro34] PYY binding in the dentate gyrus (DG). The i.c.v administration of NPYY1R agonist in a FST rat model of depression showed anti-depressive-like effect (decrease in immobility and increase in swimming), whereas the i.c.v administration of galanin caused pro-depressive behavior (increase in immobility time and decreased climbing). The antidepressant effect of NPYY1R agonist was, however, noticeably improved when galanin and NPYY1R agonist were administered together. This is mainly due to the presence of GALR2 antagonist M871 in this area [23]. 

Moreover, the implication of the galanin gene in the development of depression was studied in a group of 112 adolescents. The study showed that rs948854 minor (G) SNP located within the promoter region of the GAL gene was linked to depressive symptoms. Adolescents with the GG and AG genotype for the A/G (rs948854) SNP displayed higher depressive scores than the ones carrying homozygous AA. This study, therefore, suggests that the SNP rs948854 in the GAL gene may have a role in mediating depressive symptoms especially in adolescents with the GG genotype [90].

GAL in Anxiety

Similar to how it affects depression, GAL’s impact on anxiety is dependent on the receptor it is acting on. In contrast to the deletion of GAL2R, the loss of GAL3R led to anxiety-like symptoms in mice [91]. The intra-DRN administration of GAL1R agonist M617 improved inhibitory avoidance in ETM and OF anxiety model of rats, indicating an anxiogenic effect. However, anxiolytic-like effects were produced by the intra-DRN infusion of the GAL2R agonist AR-M1896. Neither of these agonists had a significant influence on OF’s locomotor activity or altered ETM’s escape behavior. Moreover, the prior treatment with WAY100635, a 5-HT1A antagonist, via intra-DRN infusion decreased the anxiolytic effect induced by AR-M1896 in rats tested in the ETM. This suggests that the anxiolytic effect mediated by GAL2 receptors depends on the serotonergic systems [24]. Furthermore, on an EPM anxiety model of rats the intra-dorsal hippocampal administration of GAL caused anxiogenic effects, while the administration of GAL2R antagonist M871 had anxiolytic-like effects. M871 therapy, on the other hand, prevented the GAL-induced anxiogenic effect in the dorsal hippocampus [25]. Furthermore, a study conducted on 597 subjects who handed in their DNA samples showed a significant association between anxiety and the GAL rs948854_C–rs4432027_C haplotype and the rs1042577_T single-locus allele, respectively [92]. 

### 4.2. Hypothalamic Releasing Hormones (HRH) 

The hypothalamus, located in the ventral brain, coordinates the endocrine system by receiving various signals from multiple brain regions and responding by secreting both inhibiting and releasing hormones. One of the principal efferent pathways from the hypothalamus is the hypothalamic neurohypophysial tract, which links the paraventricular and supraoptic nuclei to the nerve terminals in the median eminence, toward the anterior pituitary gland, and the posterior pituitary gland. Through the HP axis (Figure 2), the hypothalamus and pituitary gland work together to regulate body processes. The hypothalamus itself contains various neurons that release hormones and neuropeptides with multiple functions, some of which impact psychological processes and, thus, affect behavioral disorders such as depression and anxiety [93].

#### 4.2.1. Corticotrophin-Releasing Factor (CRF)

CRH is a 41-amino-acid hormone regulated by the HPA axis [13]. CRF functions as a hormone when secreted in the parvocellular neurons of the hypothalamic paraventricular nucleus. However, when it is secreted in other brain regions, it acts as a neurotransmitter. CRF mainly acts through two different receptors CRF1 and CRF2 [15]. CRF’s involvement in depression and anxiety have been demonstrated in recent studies.

CRF in Depression

The involvement of CRF in depression and depressive-like symptoms has been well demonstrated in the majority of studies. A corticotropin-releasing factor 1 receptor blocker, CP154526, and an antidepressant, fluoxetine, were each given to different groups of rats in the chronic unpredictable mild stress (CUMS) model. In a third group, both medications were given. The findings demonstrated that CP154526 improved locomotor function, decreased immobility time, and boosted sucrose preference. This suggests that CP154526 has an antidepressant-like effect. Moreover, this study found that CP154526 inhibits CRH expression in the serum of CUMS rats and downregulates the expression of BDNF and GAP43 in the hypothalamus of CUMS rats. This suggests that the antidepressant effect of CP154526 may be associated with HPA axis modulation effects such as lower serum CRH concentrations, as well as lower BDNF and GAP43 expression in the hypothalamus [13]. Moreover, the i.c.v administration of CRF in mice noted reduced immobility in TST and FST which is an anti-depressive-like behavior. However, in chronically foot-shocked mice, i.c.v administration of CRF had no similar effects. Additionally, floating was reduced in mice but increased in rats when the i.c.v. CRF was used. The responses in rats to the highly stressful environment were reversed by CRF [14]. 

Moreover, in another study conducted on a corticosterone (CORT) model of depression rats, a significant increase in CRF in the amygdala, hypothalamus, and peripheral blood was observed, accompanied by less mobility in FST after CORT treatment. The intravenous (i.v) infusion of SN003, a CRF1 receptor inhibitor, opposed the CORT-induced immobility. The effect of SN003 was similar to the effect of imipramine (a tricyclic antidepressant) and fluoxetine (a selective serotonin reuptake inhibitor) that were injected intravenously. In fact, i.v administration of (0.5 mg/kg) of SN003 enhanced the antidepressant activity of fluoxetine and imipramine in CORT-rats. Moreover, the coadministration of SN003 and the conventional antidepressants caused a decrease in CRF levels in the tested biological material, but no treatment reduced CRF levels in the amygdala, hypothalamus, and peripheral blood. However, the administration of SN003 or fluoxetine, at doses that were not enough to cause any significant behavioral effect, was able to reduce the CRF elevated levels in the amygdala, hypothalamus, and peripheral blood. These findings suggest that a CRF1 receptor blocker could be useful to treat depression and depressive behavior as its blockage causes antidepressant-like effects [15].

CRF in Anxiety

CRF was shown to also be involved in anxiety and anxiety-like symptoms. Stress-induced anxiety-like symptoms were reduced by lowering CRF in the amygdala’s central nucleus. Moreover, the reduction of CRF in the bed nucleus of the stria terminalis (BNST) also decreased stress-induced modifications in the CRF receptor expression [94]. In accordance with these findings, another study conducted on juvenile rainbow trout i.c.v injection of CRF showed mouth opening in the subjects, constant opercula flaring, and aggressive head shaking from side to side, all of which are indications of anxiety [95]. Furthermore, in a study conducted on male EPM rats, bilateral injection of CRF into the Fr2 region of the frontal cortex displayed anxiolytic-like effects. Stressin 1, a CF1R agonist, was administered, and the same outcome was seen. Administration of NBI 27914, a CF1R antagonist, however, blocked this effect [17]. 

Additionally, a study using a mouse model of mild traumatic brain injury (mTBI) anxiety revealed the contribution of CRF receptor 1 to anxiety. This model was chosen as mTBI was previously shown to be involved in the dysregulation of HPA axis, which may impact behavioral disorders. In this study, the i.c.v injection of CRF increased anxiety-like symptoms and the HPA axis response to stress. However, the i.c.v injection of CRF1R antagonist, antalarmin, reduced these responses and had an anxiolytic effect. Accordingly, these findings imply that CRF mediates anxiety and anxiety-like behavior via its CRF1 receptor [16]. The role of the CRF1 receptor was also demonstrated in another study that showed that, via CRF1R, CRF induces the enzyme fatty acid amide hydrolase (FAAH), which reduced endocannabinoid anandamide (AEA) in the amygdala, causing anxiety-like behavior. Nevertheless, it was observed that FAAH inhibition prevented this anxiety-like behavior [96]. 

#### 4.2.2. Hypocretin/Orexin

Hypothalamic neuropeptides called orexins, also referred to as hypocretin, are highly expressed in the central and peripheral nervous systems. There are two types of orexin, orexin-A (or OrxA hypocretin-1) made up of 33 amino acids, and orexin-B (OrxB or hypocretin-2) made up of 28 amino acids. Both derive from the same precursor peptide prepro-orexin [97] and are similar in structure. Orexin-A has a higher affinity to the orexin-1 receptor (OX1R), whereas orexin-2 has an equal affinity for both receptors 1 and 2 (OX2R). These receptors are coupled to G-proteins and are found on presynaptic neurons, as well as on postsynaptic neurons. Orexinergic neurons are found primarily in the lateral hypothalamus area (LHA) and posterior hypothalamus, and they project to the entire neuroaxis [98].

Orexin in Depression

Orexin has been shown to be involved in the pathophysiology of depression. Orexin levels were found to be higher in a rat FSL depression model compared to FRL healthy controls [35]. Orexin has also been shown to induce antidepressant behavior through GABAergic ventral pallidum (VP) neurons. Furthermore, the inhibition of orexin receptors, by microinjecting TCS1102, an ORX1 and ORX2 receptor antagonist, in the central pallidum elicited depressive-like behavior [36]. Additionally, a different study using an FST rat model of depression revealed a connection linking orexin distribution, its mRNA receptors, and depressive state. Results showed that, in the hippocampus, animals that showed more depressive-like behavior had a lower expression of OrxA. In the amygdala, there was a curvilinear association between OrxA and FST. However, a positive correlation was noted between Orx1 receptors and depressive behavior [98].

In accordance with these findings, in a study carried out on an unpredictable chronic mild stress (UCMS) rodent model of depression, orexinergic neurons were shown to be more activated in the perifornical and dorsomedial hypothalamic region of UCMS-affected mice compared to the lateral hypothalamus. This upregulated activation was counteracted by treatment with the antidepressant fluoxetine. UCMS further reduced the expression of ORX2 receptors in the thalamus and hypothalamus, although this reduction was not seen in mice treated with fluoxetine [99]. Moreover, on a defeat model of depression, orexin level was noted to be downgraded in the ventral tegmental area (VTA) and medial prefrontal cortex (mPFC), as well as decreased in the hypothalamus [100]. 

The involvement of orexin in depression was further demonstrated by studying the effect of Xiaoyaosan, a Chinese medicine used for emotional disorder treatment with antidepressant effects, on a rat model of depression. The depression model group had significantly lower orexin-A mRNA expression in the lateral hypothalamic area than the control group. OxR1 levels were also significantly reduced in the depression model. These levels were reduced by using Xiaoyaosan. Therefore, Xiaoyaosan can modulate the antidepressant effect by acting through orexin-A/OxR1 in the lateral hypothalamic area [101].

Orexin in Anxiety

Several studies have found orexin to be involved in anxiety. Orexin-deficient mice displayed increased anxiety in LBD, OF, and carnivore-induced avoidance tests. They were observed to be normal in terms of fear and safety learning. This shows that orexin plays a role in the manifestation of anxiety [102]. Moreover, a study conducted on 56 adolescents diagnosed with any anxiety disorder and not taking medication and 32 healthy controls showed that orexin-A levels were notably higher in the anxiety group than in healthy controls. A positive correlation was noted between anxiety traits and orexin-A [103]. Furthermore, in a study carried out on an EPM model of anxiety, microinjections of OxA and OxB in the paraventricular nucleus of the thalamus (PVT) reduced open arm time and entries, which is indicative of anxious behavior. This behavior was counteracted by the microinjection in the PVT of CRF antagonists or opioid kappa receptors. Additionally, following a foot-shock stress, microinjection of the Ox2R antagonist TCSOX229 in the PVT showed anxiogenic effects [37]. 

Moreover, i.c.v. treatment of OxA improved locomotor activity in goldfish, but this effect was reversed by i.c.v. administration of SB334867, an OX1R antagonist. The i.c.v administration of OxA in a black and white test and an upper lower test revealed that the goldfish spent less time in the white area and took longer to move from the lower to the upper area. This is consistent with an anxiogenic effect, which was also reversed by i.c.v injection of SB334867 [38]. Similarly, i.c.v. orexin in the central nucleus of hamsters used in an EPM and LBD hamster anxiety models showed anxiogenic effects (more time spent in the dark place and closed arms during EPM). However, these effects were inhibited by the administration of flunitrazepam GABA receptor 2 subunit agonists [39]. 

#### 4.2.3. Melanin-Concentrating Hormone (MCH)

MCH is a cyclic neuropeptide made up of 19 amino acids [28]. MCH-synthesizing neurons are primarily localized in the lateral hypothalamus and incerto-hypothalamic region. MCH acts through the activation of two receptors MCH receptor 1 (MCHR1) and MCH-R2 that are coupled to a G-protein. The first type of receptor is the only one found in rodents and is highly expressed in limbic brain regions such as the prefrontal cortex, nucleus accumbens, amygdala, and hippocampus [104]. MCH is thought to play a role in the pathophysiology of depression and anxiety. 

MCH in Depression

The involvement of MCH in depression is evident, as injection of a low dose of MCH in the dorsal raphe nucleus (DRN) increased immobility time and increased climbing in a forced swim test model of rats, indicating pro-depressive behavior. As a result, this depressive behavior was reduced after treatment with fluoxetine, a selective serotonin reuptake inhibitor antidepressant. Similarly, MCH immunoneutralization resulted in an antidepressant effect [26]. In a forced swim test model of rats, the depressive behavior elicited by MCH was reversed after intra-DRN administration of ATC0175, an MCH-1 receptor antagonist, or intraperitoneal pretreatment with nortriptyline, a noradrenergic antidepressant [27]. This indicates that the melanin-concentrating hormone induces a depressive behavior in the dorsal raphe nucleus. 

Likewise, rats who underwent a forced swim test and a sucrose preference test after receiving MCH in the locus coeruleus (LC) displayed depressive-like behavior. The same effect was observed was a result of ic.v. administration of MCH, or chronic subcutaneous injections of corticosterone. The MCH receptor 1 antagonist SNAP-94847, however, caused this pro-depressive behavior to change [28]. The development of depressive-like behavior is, thus, influenced by the MCHergic system in the locus coeruleus.

Nevertheless, melanin-concentrating hormone was also reported to have a therapeutic function as it displayed an antidepressant-like effect in stress models of mice and rats. This anti-depressive effect was shown to be mediated by the activation of the signaling molecules mTOR. This effect was stopped by the administration of mTOR inhibitor and MCHR1 antagonist [29]. 

MCH in Anxiety

The MCH neurons of the LHA project to the basolateral amygdala (BLA). Mice developed an anxiety disorder as a result of the chemogenetic stimulation of MCH neurons and the microinjection of MCH intra-BLA. This anxious behavior was, however, decreased by the administration of SNAP-94847, an MCHR1 antagonist. Furthermore, intra-BLA administration of MCH in a chronic acute combining stress model of mice had an anxiolytic effect by boosting anxiety-like behaviors [30]. However, the intra-DRN administration of MCH to an elevated plus-maze model of rats caused no changes in anxiety behaviors [27]. Nevertheless, the central injection of TPI 1361-17, an MCH1R antagonist with high affinity, revealed strong anxiolytic effects in mice using elevated maze and light–dark tests as models of anxiety [31]. Consequently, the MCHR system may contribute to the emergence of anxious behavior. 

#### 4.2.4. Oxytocin (OT) 

OT is a neuropeptide, made up of nine amino acids, synthesized mainly in the supraoptic nucleus (SON), paraventricular nucleus (PVN), and accessory nuclei of the mammalian hypothalamus. Numerous brain areas, including the medial amygdala, suprachiasmatic nucleus, BNST, locus coeruleus, and dorsomedial hypothalamus, contain oxytocinergic neurons. Magnocellular neurons and smaller parvocellular neurons make up the two subpopulations of neurons found in the PVN of the hypothalamus. While parvocellular OT neurons project toward brain areas such as the brainstem, spinal cord, or supraoptic nucleus to release OT in a somato-dendritic manner, magnocellular OT neurons primarily project to the neurohypophysis, where OT is secreted into the peripheral bloodstream via neurohemal contacts [105]. OT receptors are found in a variety of peripheral sites, including the adrenal and pituitary glands, as well as central receptors in the limbic area [106]. OT functions as a neurotransmitter or neuromodulator in the central nervous system [34]. The expression of OT and OT receptors has been found to be involved in the manifestation of behavioral disorders such as depression and anxiety. 

OT in Depression

In order to understand the correlation between depression and oxytocin, a study was conducted on 108 Hispanic women who were in their third trimester of pregnancy. This study showed that 28% of women had potential depression during their pregnancy, and that 23% had potential depression 6 weeks postpartum. OT levels were significantly reduced from prenatal to postpartum in all participants, with the exception of those with potential prenatal depression, who showed no significant variations in OT levels. Moreover, OT levels were significantly higher in women who were anxious or depressed 6 weeks postpartum [107]. Another study, however, discovered that mothers with persistent perinatal depression had markedly increased overall OT receptor methylation than other groups [108]. Another study assessed OT levels in a group of 40 patients (30 women and 10 men) who had been diagnosed with major depressive disorder or a bipolar affective disorder depressive episode, and then compared them to a group of 32 healthy controls (20 women and 12 men). The results showed that serum levels of OT were lower in all diagnosed patients when compared to healthy controls. Moreover, serum OT levels were not modified by the treatment with antidepressant drugs or electroconvulsive therapy. A gender difference was also observed, with female patients having significantly lower OT levels than healthy females, whereas no significant difference was observed between diagnosed males and male controls [106].

OT in Anxiety

The involvement of OT in anxiety disorders and anxious behavior has been demonstrated in multiple studies. A study was conducted on 56 male soccer players to understand the relationship between anxiety and OT level in human models, and levels of salivary OT and cortisol were measured before and after a soccer match. The results revealed that those who won had significantly less cognitive anxiety, higher self-confidence, and higher OT levels. However, those who lost had lower OT levels but higher cortisol levels [109]. 

The injection of OT into the anterior cingulate cortex (ACC) of sham and common peroneal nerve ligation mice majorly increased the mechanical withdrawal thresholds, implying a reduction in severe pain. Moreover, on the same animal model, the bilateral microinjection of OT into the ACC reversed the reduction time in open arms, number of total crossings, and total traveled distanced caused by injury. This suggests that OT may have an anxiogenic effect. In an open field test, similar results were reported using the same animal model. Atosiban, a peptide OT antagonist, and L-371,257, a nonpeptide OT antagonist, were also administered bilaterally through microinjection into the ACC of the same animal model. The administration of each alone did not have any significant changes. However, pretreatment with either, followed by the administration of OT after 20 min, showed blockage of the anxiolytic effect caused by OT in the anterior cingulate cortex [34]. 

### 4.3. Opioid Neuropeptides

Endogenous opioid peptides are molecules produced by the central nervous system, as well as by other glands, such as the adrenal and pituitary glands. These peptides have both hormonal and neuromodulatory properties. Opioid neuropeptides act as neuromodulators, modifying the electrical properties of their target neurons to modulate the actions of other neurotransmitters in the central nervous system. These neurons become more difficult to excite, allowing opioid neuropeptides to influence neurotransmitter release. As a result of these changes, opioid neuropeptides are involved in euphoria, pain relief, and psychiatric disorders, as well as influencing other behaviors such as alcohol consumption [110].

#### 4.3.1. Enkephalin (ENK)

ENKs are endogenous opioid pentapeptides produced in the adrenal medulla and central nervous system. Structurally, there are two different ENK peptides: Leu-ENK (YGGFL) and Met-ENK (YGGFM). These two peptides are derived from a precursor protein called proENK via post-translational proteolytic cleavage. In mammals, five ENK sequences are present in proENK: one copy of Leu-ENK and four copies of Met-ENK. Hence, the proENK gene’s activity is magnified when multiple ENK peptides are produced. In general, ENK peptides bind to the delta opioid receptor (DOR) and are involved in pain modulation, neurotransmission, mood regulation, neuroendocrine functions, and movement [18].

ENK in Depression

Enkephalins, in addition to their analgesic properties, play an important role in stress responses and motivated behaviors. In earlier investigations, participants given a DOR antagonist, which blocks the receptor that ENK normally binds to, exhibited depressive and anxious behaviors. These behaviors were also noted in animals lacking the precursor of ENK (proENK) or DOR [111]. The role of ENK–DOR signaling in modulating depressive behaviors has been studied in multiple brain regions. For instance, rats exposed to foot-shock stress in one study [112] and forced swimming in another study [113] exhibited lower levels of Leu-ENK in the hypothalamus and Met-ENK in the striatum and hypothalamus [113]. Additionally, chronic usage of the antidepressants imipramine and iprindole increased the levels of ENK in NAc in rats, whereas chronic mild stress decreased ENK levels [114]. Furthermore, when given enkephalinase inhibitors in the nucleus accumbens, mice displayed greater resilience to stress in a social interaction test, which was able to reduce depressive-like behaviors. This study looked at the effects of inflicting stress on mice. Overall, these earlier studies point to a critical function for ENK–DOR signaling in the regulation of depressive behaviors and behavioral responses to stress [115].

ENK in Anxiety

The central nucleus of the CeA contains high concentrations of the endogenous opioid ENK, which may play a role in the regulation of anxiety and fear. Experiments on mice have shown that, when the levels of ENK in the amygdala decrease, the fear, anxiety, and aggressiveness of mice increases. By contrast, increasing the levels of ENK or reducing its breakdown reduces such behaviors [116]. One study examined the functional role of ENKergic neurons in the BLA by modifying levels of ENK in rats. Findings demonstrated that behavioral abnormalities such as increased anxiety in social interactions occur from the depletion of ENK in the BLA [117]. In another study, a CUS stress model was administered to vulnerable individuals. It is clear that the BLA’s ENK expression was not being adequately compensated for. These findings suggest that the adaptability to anxiety and chronic stress is mediated by adaptive mechanisms that allow ENK recovery in the BLA [118]. 

#### 4.3.2. Endorphins 

Endorphins, also known as the body’s natural painkillers, were the first endogenous opioid peptides to be discovered. They are released by the pituitary gland and hypothalamus in response to stress or pain. Endorphins are also strongly linked to states of pleasure, such as those induced by love, laughter, and sex. There are three types of endorphins, with beta-endorphins being the most studied and prevalent. They are best known for their pain-relieving effect and for being associated with exercise-induced euphoria [119]. Endorphins are also known for their functional duality, as they act as hormones in the pituitary gland and as neuromodulators or neurotransmitters in the central nervous system [120]. 

Endorphins in Depression

The opioid system is important in mediating social attachment and analgesia, and it may also have an impact on depression. In general, opioid systems can signal reward by releasing endogenous opioids when motivated behavior is displayed. A study examining the brain processes of participants diagnosed with depression found that μ-opioid receptor neurotransmission increases in the anterior cingulate during induced sadness [121]. Beta-endorphin levels, in particular, can be used to diagnose depression, given that individuals with depression exhibit abnormal levels of endorphins. A study observed the effect of the administration of citalopram, an antidepressant, to depressed patients while maintaining a control group of depressed individuals who did not take the medication. Beta-endorphin levels in each group were equal at baseline. After 8 weeks of citalopram treatment, beta-endorphin levels decreased significantly compared to the control group. This indicates a strong correlation between the levels of beta-endorphins and depression [122]. Furthermore, another study examining the relationship between exercise and depression found that regular exercise promotes the release of endorphins, which in turn improved the mood of participants. This suggests that endorphins can be used in therapeutic strategies for depression [123]. 

Endorphins in Anxiety

Anxiety disorders are common psychiatric conditions that are chronic and can negatively affect multiple aspects of one’s life. The opioid system has an important role in the neural modulation of anxiety. Multiple studies found that exercise contributes to the release and binding of beta-endorphins to their receptor sites in the brain, which promotes reduced anxiety and mood elevation [124]. A study found that phobic anxiety is linked to a significant elevation in patients’ plasma beta-endorphin levels. These levels stabilized at normal levels after the anxiety subsided. This is in line with the hypothesis that the homeostatic endorphinergic response to stress and anxiety functions as an effective mechanism in prolonged exposure therapy for the treatment of phobias [125]. Short-term aerobic exercise has also been shown in multiple studies to help lower anxiety sensitivity, which is the tendency to misinterpret anxiety-related symptoms and worry that they can have severe social, physical, or psychological consequences. Individuals with high anxiety sensitivity might benefit from exposing themselves to the symptoms they fear (such as rapid heartbeat) through regular exercise, thus increasing their tolerance for such symptoms [124]. 

### 4.4. Pituitary Hormones 

The pituitary gland, an endocrine organ about the size of a pea, secretes a number of hormones that are crucial for regulating the function of other endocrine organs. These hormones include prolactin, growth hormone (GH), follicle-stimulating hormone (TSH), adrenocorticotropic hormone (ACTH), and luteinizing hormone (LH). Each of these hormones serves a distinct purpose [126]. Prolactin, for example, stimulates lactation after giving birth. ACTH is involved in the stress response by stimulating the adrenal glands to produce cortisol (the stress hormone), which has various functions such as maintaining blood pressure and regulating metabolism. FSH stimulates sperm production in males and estrogen production, as well as egg development in females. LH stimulates ovulation in females and testosterone production in males, as well as regulating the functions of the gonads. GH stimulates growth in children, while it plays a role in maintaining healthy muscles and metabolism in adults. Lastly, TSH stimulates the thyroid to produce hormones that regulate metabolism, the nervous system, and energy levels. The release of these hormones is regulated by stimulatory and inhibitory signals from the hypothalamus [127].

#### 4.4.1. Arginine-Vasopressin (AVP)

AVP also known as antidiuretic hormone (ADH), is a nine-amino-acid neuropeptide located in the hypothalamus. It regulates kidney function and water reabsorption. It also regulates blood pressure, osmotic balance, and sodium homeostasis. Several diseases are caused by a disruption in ADH secretion or levels [128]. Furthermore, AVP is thought to play an important role in the pathophysiology of affective disorders, as clinical and postmortem studies have found increased levels of vasopressin in the brain and plasma of depressed and anxious patients [129]. 

AVP in Depression

Because it functions as a neuromodulator of the stress response, AVP is heavily involved in mood disorders. For example, AVP levels increase significantly in healthy subjects when they are exposed to severe psychological stressors [129]. Moreover, AVP binds to V1b receptors in the anterior pituitary, which are involved in the activation of the HPA axis. The HPA axis is found to be more active in subjects diagnosed with major depressive disorder, which implies a higher rate of synthesis and release of AVP, driving the axis in such cases instead of CRH. Additionally, patients with major depressive disorder (MDD) have a substantial increase in AVP neurons and V1b receptors [130]. One study enrolled 52 patients with MDD, of whom 18 were hospitalized and 43 were outpatients, to compare AVP levels in healthy and depressed patients. Researchers found that depressed patients had significantly higher concentrations of AVP compared to healthy individuals [131]. 

AVP in Anxiety

AVP seems to be involved in daily anxiety-related behavior, as well as anxiety disorders. It is proposed that, if AVP’s central release patterns reach an upper limit of a continuum, “normal” anxiety might progress to pathological anxiety, with AVP aiding in this psychopathological process. Additionally, it is believed that genetic susceptibility and dysregulated stress responses contribute to psychopathology. AVP has been shown to play a role in genetic expression in addition to responding to stressful stimuli. In the hypothalamic paraventricular nucleus, an excess of AVP caused by polymorphism was clearly linked to a severe anxiety phenotype [132]. One study found that the AVP receptor V1aR plays a role in regulating anxiety-related behavior and social recognition in mice. Subjects were exposed to stressful and anxiety-inducing stimuli such as an elevated maze, forced swimming, and a lit open-topped box connected to a dark closed box. Mice tended to spend more time in open areas that were well lit, and they exhibited anxious tendencies when forced to do tasks under pressure or in less illuminated areas. An increase in AVP levels and activation of the V1aR have been linked to increased anxiety in mice [133]. 

#### 4.4.2. Adrenocorticotropic Hormone (ACTH)

ACTH is a tropic hormone that regulates cortisol and androgen production. It is produced by the pituitary gland and controlled by the HPA axis. It is associated with many diseases including Addison disease, Cushing disease, and affective disorders such as depression and anxiety [134]. 

ACTH in Depression

Because of its effect on the release of the stress hormone cortisol, ACTH has been shown to be extensively involved in mood disorders such as depression. One study compared ACTH levels with the HPA axis. Activity in depressed and healthy subjects found that MDD patients had significantly higher levels of ACTH compared to their healthy counterparts, as well as irregular HPA axis activity [135]. Another study that looked at HPA axis activity during depression enrolled 43 patients with MDD. After 6 weeks of treatments with the antidepressant fluoxetine, ACTH and cortisol levels decreased significantly compared to baseline. Hence, it was postulated that the initial recovery of the HPA axis is mediated by the return of glucocorticoid negative feedback on ACTH levels [136]. 

ACTH in Anxiety

Many studies have established a role between anxiety and the release of ACTH. In one study, rat pups were separated from their mothers for 3 h each day for 14 days to examine the effects on adult behavior. Researchers found that maternal separation led to increased ACTH and cortisol levels, as well as abnormal and anxiety-related behaviors in adulthood [137]. According to results of another study, the anxiogenic effects of ACTH peaked while individuals were performing a stressful task. However, it was also shown that chronic administration of the anxiolytic drug chlordiazepoxide decreased ACTH levels, proposing that anxious behaviors result from the action of ACTH in the midbrain and hypothalamus [137,138].

## 5. Conclusions

This review discussed the compelling evidence that neuropeptides play a role in the pathophysiology and development of behavioral disorders, specifically anxiety and depression. The effect of neuropeptides on behavior was studied using various animal models to induce depressive behavior (olfactory bulbectomy, forced swim test or Porsolt swim test, Flinder’s sensitive line, and tail suspension test), anxious behavior (Vogel’s conflict test, elevated plus maze, open field test, and light–dark box test), or stressful behavior (single prolonged stress). The effect of a neuropeptide on anxiety and depression depends greatly on the neuropeptide, its receptors, and the regions of the brain it acts on. Some exhibit pro-depressive behavior when acting through one receptor and anti-depressive behavior when acting through another (NPY). However, all of the neuropeptides listed in this article, including gut–brain peptides, hypothalamic releasing hormones, opioid neuropeptides, and pituitary hormones, have an effect on depressive and anxious behavior. As a result, this review should serve as an impetus to expand existing research on the impact of neuropeptides and their involvement in behavioral disorders. Additionally, this can serve as a foundation for additional investigation into the therapeutic benefits of neuropeptides in the treatment of depressed and anxiety disorders.

## Figures and Tables

**Figure 1 bioengineering-10-00258-f001:**
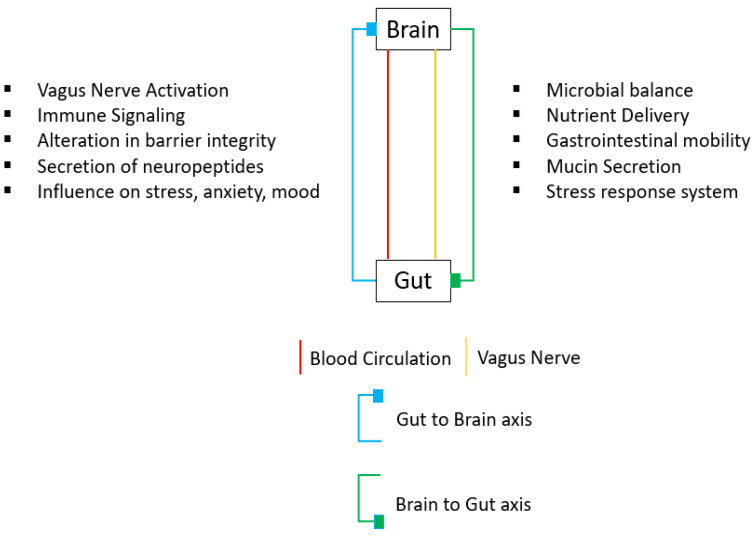
**Gut–brain axis.** This is a bidirectional axis between the brain and the gut, connected by blood circulation (shown in red) and the vagus nerve (shown in yellow). Through the gut-to-brain axis (shown in blue), the vagus nerve gets activated, the barrier’s integrity gets altered, and neuropeptides and immune signaling molecules are secrete, with an effect on stress and anxiety. Through the brain-to-gut axis (shown in green), the main functions are microbial balance, nutrient delivery, gastrointestinal mobility, mucin secretion, and the stress response system.

**Figure 2 bioengineering-10-00258-f002:**
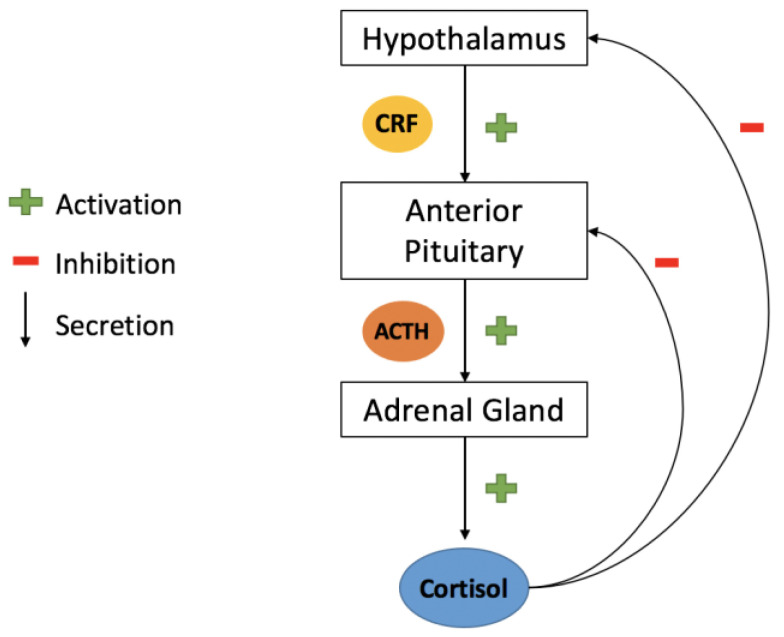
**Hypothalamic–pituitary–adrenal (HPA) axis.** Upon activation (by a stimulus), the hypothalamus releases the corticotrophin-releasing factor (CRF, in yellow), stimulating the anterior gland, which then secretes the adrenocorticotropic hormone (ACTH, in orange). The ACTH, in turn, activates the adrenal gland, which secretes cortisol (in blue). Cortisol inhibits the anterior pituitary gland, as well as the hypothalamus.

**Table 1 bioengineering-10-00258-t001:** Summary table classifying studies showing the effect of neuropeptides on depression and anxiety by model, species, and route of administration. 0: no effect; ↑: pro-depressive effect; ↓: antidepressant effect; +: anxiogenic effect; −: anxiolytic effect; SPS: single prolonged stress; FST: forced swimming test; CeA: central nucleus of amygdala; VCT: Vogel’s conflict test; CMS: chronic mild stress; EPM: elevated plus maze; OFT, OPF: open field test, OB: olfactory bulbectomy; TST: tail suspension test; NPY: neuropeptide Y; SP: substance P; NT: neurotensin; GAL: galanin; CRF: corticotrophin-releasing factor; OX: orexin; MCH: melanin-concentrating hormone; OT: oxytocin; ENK: enkephalin.

Target Neuropeptide	Administered Substance	Model	Species	Administration	Effect	Reference
CRF	CP1544526 (CRF1 receptor blocker)	CUMS	Rat		↓	[13]
CRF	CRF	TST, FST	Mouse	i.c.v	↓	[14]
CRF	CRF	FST	Rat	i.c.v	↑	[15]
CRF	CRF	mTBI	Mouse	i.c.v	+	[16]
CRF	CF1R antagonist, antalarmin	mTBI	Mouse	i.c.v	−	[16]
CRF	CRF, stressin 1 (CF1R agonist)	EPM	Rat	Bilateral injection	+	[17]
CRF	NBI 27914 (CF1R antagonist)	EPM	Rat	Bilateral injection	−	[17]
ENK	Enkephalinase inhibitors	Social Interaction Test	Mouse	Intra-NC	↓	[18]
GAL	Galanin, GALR1 agonist (M617), GALR2 antagonist (M817)	FST	Rat	i.c.v	↑	[19]
GAL	GALR2 agonist AR-M1896	FST	Rat	i.c.v	↓	[19]
GAL	Galanin, AR-M1896	FST	Rat	Intra-DRN	↓	[20]
GAL	M617	FST	Rat	Intra-DRN	0	[20]
GAL	AR-M1896, M617	OFT	Rat	Intra-DRN	0	[20]
GAL	GAL (1-15)	OB	Rat	i.c.v	↓	[21]
GAL	M871	OB	Rat	i.c.v	↑	[21]
GAL	GalR2 selective agonist M1160	TST	Mouse	i.c.v	↓	[22]
GAL	Galanin + NPYY1R agonist	FST	Rat	i.c.v	↓	[23]
GAL	M617	ETM, OF	Rat	Intra-DRN	+	[24]
GAL	AR-M1896	ETM, OF	Rat	Intra-DRN	−	[24]
GAL	Galanin	EPM	Rat	Intra-dorsal hippocampus	+	[25]
GAL	M871	EPM	Rat	Intra-dorsal hippocampus	−	[25]
MCH	MCH	FST	Rat	Intra-DRN	↑	[26]
MCH	MCH-1 receptor antagonist ATC0175	FST	Rat	Intra-DRN	↓	[27]
MCH	MCH	FST	Rat	Intra-LC	↑	[28]
MCH	SNAP-94847, a MCH receptor 1 antagonist	FST	Rat	Intra-LC	↓	[28]
MCH	MCH	Stress model	Rat, Mouse	Intranasal	↓	[29]
MCH	MCH	EPM	Mouse	Intra-BLA	+	[30]
MCH	SNAP-94847, a MCH receptor 1 antagonist	EPM	Mouse	Intra-BLA	−	[30]
MCH	MCH	Chronic acute combining stress	Mouse	Intra-BLA	−	[30]
MCH	TPI 1361-17, a MCH1R antagonist	EPM, LBD	Mouse	Central administration	−	[31]
MCH	MCH	EPM	Rat	Intra-DRN	0	[27]
NT	NT1R antagonist (SR 48,692) + NT after 15 min	OPF	Rat	Bilateral micro- injection in the VP	+	[32]
NT	NTS1 agonist, NT	EPM, LBD, OFT	Rat	Injection into the PrL	+	[33]
NT	NTS1 antagonist	EPM, LBD, OFT	Rat	Injection into the PrL	0	[33]
OT	OT	Common peroneal nerve (CPN) ligation	Mouse	Intra-ACC	−	[34]
OX	OX	FRL	Rat	Microinjection in VP	↓	[35]
OX	TCS1102, ORX1 and ORX2 receptor antagonist	FRL	Rat	Microinjection in VP	↑	[36]
OX	OxA, OxB	EPM	Rat	Microinjection into the PVT	+	[37]
OX	TCSOX229 (Ox2R antagonist)	EPM	Rat	Microinjection into the PVT	−	[37]
OX	OxA	Black and white test	Goldfish	i.c.v	−	[38]
OX	SB334867 (OX1R antagonist)	Black and white test	Goldfish	i.c.v	+	[38]
OX	OX	EPM, LBD	Hamster	i.c.v	−	[39]
OX	Flunitrazepam (GABA receptor 2 subunit agonist)	EPM, LBD	Hamster	i.c.v	+	[39]

## Data Availability

Not applicable.

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
