# Peer review of "Protective Role and Functional Engineering of Neuropeptides in Depression and Anxiety: An Overview"

_bioengineering, 2023, doi:10.3390/bioengineering10020258_

Round 1
Reviewer 1 Report
This review article presents and discusses the potential roles or various neuropeptides and peptide hormones in anxiety and depression disorders. The scope of the article is reasonably comprehensive, and the organization is clear. There are a few queries that need to be addressed.
1) p1,L43 - 'which weighs down the society as a whole'. This is a repetitive point, as the authors have already stated in line 36 that 'depression and anxiety place a heavy strain on themselves, their families, and society '. To labour the point could come across as victim-blaming.
2) Table 1 is useful. However, what is the rationale for the organization of the table? It's not alphabetical, or anatomical (as far as I can work out). It would be easier to access the information in the table if it were organized in a more logical order.
3) Perhaps this is because the authors do not classify them as neuropeptides, but natriuretic peptides have also been shown to have anxiolytic effects (and these peptides - CNP in particular - are expressed in and released from multiple parts of the CNS). Suggest looking at: Medvedev A et al., 2005 (PMID 15854583); Jahn H et al, 2001 (PMID 11222988); Grade et al, 1997 (PMID 9180365); Biro et al, 1996 (PMID 8868301).
4) Whilst Section 3 is useful to those who have no experience with testing/modeling depressive disorders, does the section need to be as long as it is? Could this be abridged and appropriate referencing put in place instead?
5) p16, L539 - 'Rats' responses' - this is awkward grammar. I suggest Re-wording to something like 'The responses in rats to the highly stressful environment were reversed by CRF'.
6) p16 L548 - there is an extraneous hyphen 'the-'
7) p22. Oxytocin is covered in the previous section, but as it too, like AVP, originates from the hypothalamus but can be secreted from the posterior pituitary, perhaps there should be a brief section in the pituitary hormones paragraphs about oxytocin. Patients with hypopituitarism, or who have suffered pituitary trauma, often show social, behavioral and anxiety changes, potentially as a consequence of reduced OT.
8) p23 L913 'undeniable' is overstating the evidence - everything can be denied or refuted! Suggest using 'compelling' instead.
9) There a multiple problems with the formatting of the references; I have found inconsistencies with the following: 18, 19, 21, 25, 28,29, 31, 32, 35, 39, 41, 47, 48, 56, 57, 60, 62, 64, 66-69, 74, 92, 93, 121, 126, 127, 129, 133, 138. Authors names are missing, the use of 'et al' in author lists is not consistent. Suggest that the authors very carefully check and correct the whole reference list.
Author Response
This review article presents and discusses the potential roles or various neuropeptides and peptide hormones in anxiety and depression disorders. The scope of the article is reasonably comprehensive, and the organization is clear. There are a few queries that need to be addressed.
Dear reviewer, thank you very much for your opinion and insightful comments.
1) p1,L43 - 'which weighs down the society as a whole'. This is a repetitive point, as the authors have already stated in line 36 that 'depression and anxiety place a heavy strain on themselves, their families, and society '. To labour the point could come across as victim-blaming
Concerning this point, effectively you are right. Both sentences were removed.
2) Table 1 is useful. However, what is the rationale for the organization of the table? It's not alphabetical, or anatomical (as far as I can work out). It would be easier to access the information in the table if it were organized in a more logical order.
You are right, the table is now reorganized in alphabetical order.
3) Perhaps this is because the authors do not classify them as neuropeptides, but natriuretic peptides have also been shown to have anxiolytic effects (and these peptides - CNP in particular - are expressed in and released from multiple parts of the CNS). Suggest looking at: Medvedev A et al., 2005 (PMID 15854583); Jahn H et al, 2001 (PMID 11222988); Grade et al, 1997 (PMID 9180365); Biro et al, 1996 (PMID 8868301).
You are perfectly right, but indeed we preferred to discuss only neuropeptides secreted and released at the level of neurons. Natriuretic peptide and others can also act at the cerebral level but this will lead to a much more extensive description and is outside the scope of our review. But we will indeed take your comment in consideration for our future bibliographic work in this field.
4) Whilst Section 3 is useful to those who have no experience with testing/modeling depressive disorders, does the section need to be as long as it is? Could this be abridged and appropriate referencing put in place instead?
Done, we tried to make it shorter, and changed the organization as also requested by another reviewer. Thank you for this comment
5) p16, L539 - 'Rats' responses' - this is awkward grammar. I suggest Re-wording to something like 'The responses in rats to the highly stressful environment were reversed by CRF'.
Done, it’s now corrected
6) p16 L548 - there is an extraneous hyphen 'the-'
Done, it’s now corrected
7) p22. Oxytocin is covered in the previous section, but as it too, like AVP, originates from the hypothalamus but can be secreted from the posterior pituitary, perhaps there should be a brief section in the pituitary hormones paragraphs about oxytocin. Patients with hypopituitarism, or who have suffered pituitary trauma, often show social, behavioral and anxiety changes, potentially as a consequence of reduced OT.
We have already written a very complete and detailed paragraph concerning oxytocin in our manuscript, where we have also discussed as you propose the primary role of this neuropeptide. Thus, we prefer not to be very redundant. Thank you again for this relevant comment as well.
8) p23 L913 'undeniable' is overstating the evidence - everything can be denied or refuted! Suggest using 'compelling' instead.
Done, it’s now fixed
9) There a multiple problems with the formatting of the references; I have found inconsistencies with the following: 18, 19, 21, 25, 28,29, 31, 32, 35, 39, 41, 47, 48, 56, 57, 60, 62, 64, 66-69, 74, 92, 93, 121, 126, 127, 129, 133, 138. Authors names are missing, the use of 'et al' in author lists is not consistent. Suggest that the authors very carefully check and correct the whole reference list.
Thank you for this comment, you are right. We fixed the above mentioned refrences and corrected the entire references list.

Reviewer 2 Report
This manuscript demonstrated the neuropeptides' behavioral role and their functional engineering in depression and anxiety. The data suggested that neuropeptide is a promising therapeutic target in anxiety and depression. In recent years, many studies have measured the actions of drug candidates that affect neuropeptide function through targeted receptors in mood-related disorders. The research topic is of importance, while several concerns still need to be addressed.
1. The author failed to compare the distribution of different neuropeptides in the different brain areas and the relationship with depression or anxiety disorder.
2. The author didn’t mention the role of a combination of two or more neuropeptides. How did the neuropeptides change in depression or anxiety? Did the neuropeptides analogue have the same effect in mood-related disorders? whether these neuropeptides exert common downstream effects on neural systems that mediate stress. Did the neuropeptides specific to one particular disorder? The author also needs to illustrate these questions well.
3. The neuropeptide system is involved in complicated stress-related disorders, but neuropeptides may also need higher levels of stress and challenges to show significant behavioural effects in various tests. The author should also explain the doses in neuropeptide treatment, and compare it with the classical anti-depressant drug or anti-anxiety drug.
Author Response
This manuscript demonstrated the neuropeptides' behavioral role and their functional engineering in depression and anxiety. The data suggested that neuropeptide is a promising therapeutic target in anxiety and depression. In recent years, many studies have measured the actions of drug candidates that affect neuropeptide function through targeted receptors in mood-related disorders. The research topic is of importance, while several concerns still need to be addressed.
Dear reviewer, thank you very much for your opinion and your insightful comments.
- The author failed to compare the distribution of different neuropeptides in the different brain areas and the relationship with depression or anxiety disorder.
Our aim was to present the neuropeptides involved in depression and anxiety. We described the neuropeptides involved and their brain functions as well as their implication in the dysfunctions associated with certain neuronal pathologies such as depression and anxiety. Talking about their distribution, their coordination and their multiple actions leads us to make the review much denser. But this could also be the subject of another review in the future. We thank you very much for this pertinent comment.
- The author didn’t mention the role of a combination of two or more neuropeptides. How did the neuropeptides change in depression or anxiety? Did the neuropeptides analogue have the same effect in mood-related disorders? whether these neuropeptides exert common downstream effects on neural systems that mediate stress. Did the neuropeptides specific to one particular disorder? The author also needs to illustrate these questions well.
Throughout the review, we showed the result of the neuropeptides’ action on behavioral disorders per targeted area. It can’t be summed up in one section as each action of each neuropeptide depends on the brain area. The conclusion touched up on the overall role of neuropeptides in the pathophysiology of depression and anxiety.
- The neuropeptide system is involved in complicated stress-related disorders, but neuropeptides may also need higher levels of stress and challenges to show significant behavioural effects in various tests. The author should also explain the doses in neuropeptide treatment, and compare it with the classical anti-depressant drug or anti-anxiety drug.
Doses were mentioned in respect of the research found. Not all studies have been counter-compared 1-1 with the classical treatment in terms of doses. Those who were, were mentioned.

Reviewer 3 Report
Abstract is too general
Lines 23-24: “This review highlights the majority of the findings demonstrating 23 neuropeptides' behavioral role and functional engineering in depression and anxiety”. What does “functional engineering'' mean?
Line 39 “complicated behavioral disorder” or “complex”
The introduction is very weak. Try to give a brief depiction of the core symptoms of each mood disorder and then more in depth justify their relationship with neuropeptides. Have in mind that a role for neuropeptides in mood disorders, specially in anxiety has been somewhat controversial. The monoaminergic theories of mood disorders have dominated the neuropsychopharmacological landscape of these mental conditions. Wonder what makes neuropeptides so special as a therapeutic target.
Section 3: Preclinical 8animal models). Please begin with the anxiety-like models (LD ox, EPM, OF etc.) and then continue with the depression-like test. How about Learned Helpless? I miss it.
Lines 195-199: I miss the fact that the radius of action of neuropeptides extends far beyond the synaptic cleft (Not sure if H. Burbach mentioned that fact).
Line 200: Perhaps neuropeptides biosynthesis is simple, but it is not the case of their degradation and reabsorption. Please comment on how their action ended.
Gut-brain peptides? Do you mean gastrointestinal neuropeptides?
There is an unforgivable mistake in the review. I do not see mentioned anywhere in the text the most abundant neuropeptide in the brain, that is cholecystokinin or CCK which is involved in anxiety and to less extension in depression an suicide behaviors.
The study lacks lacked both depth and breadth.
Author Response
- Abstract is too general
Indeed, neuropeptides involved in neuronal pathologies and detailed in our review are numerous and their function is also diversified and complex. We are obliged to keep the abstract general since we cannot go into detail in the description of these molecules, which leads to a very long and not very comprehensible abstract. Nevertheless, the abstract, as it is, is very attractive and incites the reader to read the whole manuscript.
- Lines 23-24: “This review highlights the majority of the findings demonstrating 23 neuropeptides' behavioral role and functional engineering in depression and anxiety”. What does “functional engineering'' mean?
Advances in genetic engineering may allow the manipulation of peptides, including neuropeptides, for therapeutic purposes. Therefore, research on potential functional therapeutic applications of neuropeptides in a variety of disorders will expand knowledge beyond the modulation of neuronal circuits by neurotransmitters. Thus, our goal in this review is to provide the reader with a more general overview of the diverse field of neuropeptides, their structure and function, thus offering new perspectives on bio-engineering of these molecules for therapeutic applications.
Line 39 “complicated behavioral disorder” or “complex”
Done, replaced complicated with complex.
- The introduction is very weak. Try to give a brief depiction of the core symptoms of each mood disorder and then more in depth justify their relationship with neuropeptides. Have in mind that a role for neuropeptides in mood disorders, specially in anxiety has been somewhat controversial. The monoaminergic theories of mood disorders have dominated the neuropsychopharmacological landscape of these mental conditions. Wonder what makes neuropeptides so special as a therapeutic target.
Thank you for this comment, we edited the introduction taking what you said in consideration.
- Section 3: Preclinical 8animal models). Please begin with the anxiety-like models (LD ox, EPM, OF etc.) and then continue with the depression-like test. How about Learned Helpless? I miss it.
Thank you for this comment, it is now fixed.
- Line 200: Perhaps neuropeptides biosynthesis is simple, but it is not the case of their degradation and reabsorption. Please comment on how their action ended.
Thank you for this comment, we added how neuropeptides’ action ends.
- Gut-brain peptides? Do you mean gastrointestinal neuropeptides?
Yes
- There is an unforgivable mistake in the review. I do not see mentioned anywhere in the text the most abundant neuropeptide in the brain, that is cholecystokinin or CCK which is involved in anxiety and to less extension in depression an suicide behaviors.
You are right, Cholecystokinin is effectively involved in anxiety but in our review, we preferred to discuss only neuropeptides secreted and released at the level of neurons. Cholecystokinin and others can also act at the cerebral level but this will lead us to a much more extensive description and is outside the scope of our review. But we will indeed take your comment in consideration for our future bibliographic work in this field.
- The study lacks lacked both depth and breadth.
Ok, we'll take this comment into account for our future fieldwork.

Round 2
Reviewer 1 Report
The authors have made many of the suggested edits, and have answered the other queries acceptably. The reference list remains a problem, however (although it is much improved). Specifically, there are formatting inconsistencies with the following references:
21, 25, 28, 31, 35, 39, 41, 47, 64, 68, 105, 107, 108, 118, 126, 129, 131, 133, 138
Author Response
The authors have made many of the suggested edits, and have answered the other queries acceptably. The reference list remains a problem, however (although it is much improved). Specifically, there are formatting inconsistencies with the following references:
21, 25, 28, 31, 35, 39, 41, 47, 64, 68, 105, 107, 108, 118, 126, 129, 131, 133, 138
We thank you for taking the time to review our work a second time and provide feedback.
Concerning the mentioned references: all have been modified and formatted as required.
See manuscript revised: references highlighted in yellow

Reviewer 3 Report
The work still shows a lack of depth. I does not answer my main concern. If the neuropeptides play a complex neuromodulatory role of the main biogenic neurotransmitters, why the preclinical research of their pharmacology potential is so frustrating. Except for endogenous opioids, why do the authors still believe neuropetides are so important? What does really make them so special? And most importantly, the authors did not even mention the most abundant neuropeptide in the brain, that is cholecystokinin.
Author Response
The work still shows a lack of depth. It does not answer my main concern. Thank you for taking the time to review our work and provide feedback. We appreciate your insights and value your input. With regard to your comment that the work still shows a lack of depth, we understand your concerns. We recognize that there may always be room for further exploration and we strive to continually improve the depth of our future work.
We would like to emphasize that we explored within the limits of the scientific information available for our study and our goal was to provide good quality information for future research in this area.
If the neuropeptides play a complex neuromodulatory role of the main biogenic neurotransmitters, why the preclinical research of their pharmacology potential is so frustrating.
In this work, we perform a literature review of the functional roles and importance of neuropeptides in depression and anxiety without addressing the preclinical and/or clinical applications of these molecules. However, we thank you for your comment which could indeed be one of our main focus areas in writing future reviews discussing the applied development of these neuropeptides.
Except for endogenous opioids, why do the authors still believe neuropetides are so important? What does really make them so special?
The main goal of our review is to reference the literature on neuropeptides and gather all the established scientific data regarding their protective role and functionality in certain neurological pathologies such as depression and anxiety. At no point, we exposed ideas that neuropeptides are therapeutic molecules and/or drugs for treating neurodegenerative diseases.
And most importantly, the authors did not even mention the most abundant neuropeptide in the brain, that is cholecystokinin.
You are right, Cholecystokinin is effectively involved in anxiety but in our review, we preferred to discuss only neuropeptides secreted and released at the level of neurons. Cholecystokinin and others can also act at the cerebral level but this will lead us to a much more extensive description and is outside the scope of our review. But we will indeed take your comment into consideration for our future bibliographic work in this field.
